# Does Influenza Vaccination during Pregnancy Have Effects on Non-Influenza Infectious Morbidity? A Systematic Review and Meta-Analysis of Randomised Controlled Trials

**DOI:** 10.3390/vaccines9121452

**Published:** 2021-12-08

**Authors:** Katrine Pedersbæk Hansen, Christine Stabell Benn, Thomas Aamand, Martin Buus, Isaquel da Silva, Peter Aaby, Ane Bærent Fisker, Sanne Marie Thysen

**Affiliations:** 1Bandim Health Project, OPEN, Department of Clinical Research, Odense University Hospital/University of Southern Denmark, Studiestræde 6, 1455 Copenhagen, Denmark; kpedersbaek@gmail.com (K.P.H.); thomas.aamand@post.au.dk (T.A.); martin.kindberg.buus@post.au.dk (M.B.); p.aaby@bandim.org (P.A.); afisker@health.sdu.dk (A.B.F.); d109381@dadlnet.dk (S.M.T.); 2Danish Institute of Advanced Science, University of Southern Denmark, Fioniavej 34, 5230 Odense, Denmark; 3Bandim Health Project, Indepth Network, Apartado 861, Bissau 1004, Guinea-Bissau; i.silva@bandim.org; 4Department of Public Health, Aarhus University, Bartholins Allé 2, 8000 Aarhus, Denmark

**Keywords:** non-specific effects, vaccination, immune system, influenza vaccine, pregnancy, all-cause mortality

## Abstract

The recommendation to provide inactivated influenza vaccine (IIV) to pregnant women is based on observed protection against influenza-related morbidity in mother and infant. Non-live vaccines may have non-specific effects (NSEs), increasing the risk of non-targeted infections in females. We reviewed the evidence from available randomised controlled trials (RCTs) of IIV to pregnant women, to assess whether IIV may have NSEs. Four RCTs, all conducted in low- and middle-income settings, were identified. We extracted information on all-cause and infectious mortality and adverse events in women and their infants. We conducted meta-analyses providing risk ratios (RR). The meta-analysis for maternal all-cause mortality provided a RR of 1.48 (95% CI = 0.52–4.16). The estimates for miscarriage/stillbirth and infant all-cause mortality up to 6 months of age were 1.06 (0.78–1.44) and 1.11 (0.87–1.41), respectively. IIV was associated with a higher risk of non-influenza infectious adverse events, with meta-estimates of 2.01 (1.15–3.50) in women and 1.36 (1.12–1.67) in infants up to 6 months of age. Thus, following a pattern seen for other non-live vaccines, IIV was associated with a higher risk of non-influenza infectious adverse events. To ensure that scarce resources are used well, and no harm is inflicted, further RCTs are warranted.

## 1. Introduction

Influenza infection in pregnancy has been associated with increased maternal morbidity and possible adverse neonatal outcomes. Infants are particularly susceptible to influenza in early life [1]. During the first months of life, maternal antibodies offer some protection to the newborn [2]. Thus, influenza vaccination of pregnant women can be a way to increase protection against influenza in the newborn. The WHO now recommends influenza vaccination of pregnant women using inactivated influenza vaccine (IIV) [3].

Several reviews and meta-analyses have been conducted to assess the effects of influenza vaccination during pregnancy [4,5,6,7,8]. It has been concluded that influenza vaccination of pregnant women was safe [6,7,8] and significantly reduced maternal and infant influenza risk [4,8] as well as the prevalence of stillbirth [6], preterm, and low-birth-weight infants [5]. However, these reviews were, to a large extent, based on observational studies; and as observational studies inherently risk healthy vaccine bias and residual confounding, potential harms could be underestimated [7].

Particularly, it is increasingly acknowledged that vaccines, apart from their specific disease-protective effects, also have important non-specific effects [9]. The current system for testing vaccines is not set up to assess non-specific effects, as it usually focusses on pathogen-specific effects [10]. In 2014, the WHO reviewed the literature for three vaccines in the routine childhood vaccination programme, BCG, measles-containing vaccine, and diphtheria–tetanus–pertussis (DTP)-containing vaccines, and concluded that further research into the non-specific and overall health effects of vaccines was warranted [11,12].

The pattern observed so far has been that live vaccines such as BCG and measles-containing vaccines have beneficial non-specific effects, reducing all-cause mortality more than explained by their specific protective effects [12,13]. Worryingly, in observational studies, non-live vaccines such as DTP vaccines, in spite of their well-known protective effects against the targeted diseases, have been associated with higher all-cause mortality, particularly for females [13,14,15,16,17]. The mortality has primarily been from infectious diseases, suggesting that non-live vaccines, while protecting against the target infection, increase the susceptibility to other infections. The net results, observed now for six non-live vaccines, are that overall mortality and morbidity are increased among vaccinated versus unvaccinated females [13]. Specifically, with respect to IIV, studies in Hong Kong [18] and Guinea-Bissau [19,20,21] reported that IIV to children was associated with higher all-cause mortality and morbidity.

It was recently pointed out that there were several safety signals with respect to overall mortality and morbidity in the RCTs of IIV to pregnant women, which could indicate that IIV has negative non-specific effects on the risk of other serious infections in these women and their offspring. No formal meta-analysis was conducted [22]. A pooled analysis of two RCTs of IIV to pregnant women with respect to maternal and child all-cause mortality was subsequently published; it showed no effect of IIV on these outcomes [23].

Here, we review the evidence from the RCTs to assess the overall health effect of IIV on pregnant women, with a focus on assessing whether IIV may have non-specific effects on the risk of non-influenza infections in mother and child.

## 2. Materials and Methods

### 2.1. Search Strategy

We searched PubMed and Embase for RCTs published before 30 December 2020 using the search terms related to ‘influenza vaccines’ AND ‘pregnancy’ AND (‘pregnancy outcome’ OR ‘immunity, heterologous’) AND ‘randomised trial’ (Appendix A). A complete list of the search terms used can be found in Under Appendix A. We also manually searched references that the selected trials cited. Two reviewers (T.A. and M.B.) independently screened publications from the search results by titles, abstracts, and full text. At each level, irrelevant publications were excluded. In case of disagreement, a third reviewer (S.T.) was consulted. K.P.H. screened the citations in all the selected publications. Given the a priori knowledge that very few RCTs had been conducted, we did not write or register a formal protocol. The risk of bias was assessed by using the Cochrane Risk of Bias Tool template.

### 2.2. Inclusion and Exclusion Criteria

Studies meeting the following criteria were included:–RCTs assessing the effect of an IIV vs. either placebo or another non-influenza vaccine administered during pregnancy;–RCTs that contained information about one or more of the following outcomes: miscarriage, stillbirth, maternal death, infant death, maternal non-influenza infectious adverse events, and child non-influenza infectious adverse events.

When results from the same trial were reported in more than one publication, we compared the numbers for consistency, and each trial was only included once.

### 2.3. Data Extraction

Our search identified four RCTs [24,25,26,27]. For each trial, two of the authors (C.S.B. and K.P.H.) extracted information from all available publications and Appendix A. We extracted information on year of publication, geographic origin, study participants, clinical trial register ID, and type of influenza vaccine. We defined the following outcomes:(1)Maternal all-cause mortality (excluding accidents and suicide). One trial reported no maternal deaths [24]. From the other trials, deaths were either presented in the text [25,26] or in the appendix [21,27]. We excluded one death due to suicide [25] as well as two deaths due to electrocution and ‘cervical fracture, post-trauma’ [26].(2)Maternal mortality from presumed infectious causes. Causes of death were either presented in the text [25,26] or in the appendix [27]. We excluded deaths related to haemorrhage, cancers, cardiovascular events, and TB (presumably acquired prior to trial vaccination) but included death due to infection after caesarean section [27].(3)Maternal non-influenza infectious adverse events. This outcome was recorded as hospital admission in the South African trial [27], as serious adverse events in the Bangladesh and Mali trials [24,26]; the Nepal trial did not report this outcome [25]. We did not include TB and HIV, as we assumed that these were acquired before enrolment. Both the South African and Mali trials reported categories of non-mutually exclusive outcomes, so one individual could account for several outcomes, but based on the reported data from South Africa, the number of outcomes was not much larger than the number of individuals (316 all-cause hospitalisations in 304 women; 313 all-cause hospitalisations in 289 infants).(4)Miscarriage/stillbirths. The outcome was extracted from the texts [25], tables [27], and appendices [24,26].(5)Infant all-cause mortality (up to 6 months of age, excluding accidents, including influenza). The outcome was extracted from the tables [27] and appendices [24,25,26].(6)Infant mortality (up to 6 months of age) from presumed infectious causes based on the classification in the trial paper. Causes of death were presented in the appendix [26,27].(7)Infant non-influenza infectious adverse events. The outcome was extracted from the tables [27] and appendices [24,26]. Both South African and Mali trials reported categories of non-mutually exclusive outcomes, but the number of individuals with several events was limited. We excluded ‘neonatal sepsis, within 3 days of birth’, which we interpreted as included in ‘neonatal sepsis, within 28 days of birth’; otherwise, we counted each event as one event.

The information about the source of data for each outcome is presented in more detail in Appendix A.

### 2.4. Statistical Analyses

As the denominator, for simplicity, we chose a number of randomised pregnant women included in the RCTs for all outcomes. We conducted a meta-analysis using Review Manager 5.3 (The Nordic Cochrane Centre, The Cochrane Collaboration, 2014, Copenhagen, Denmark). Meta-estimates are presented as relative risks (RRs) with 95% confidence intervals (CIs). We used fixed-effect estimates, as there were fewer than five studies, and they were quite similar in their design and outcomes. For each outcome, we made funnel plots for visual assessment of systematic heterogeneity; none of the plots indicated systematic bias.

## 3. Results

We identified four RCTs (Table 1) published between 2008 and 2017. They were conducted in low-resource settings in Bangladesh, Mali, and Nepal, and in poor settlements in South Africa. Study population sizes varied from 340 to 4193 pregnant women. The South African trial included a small cohort of HIV-positive women; results were presented separately for HIV-positive (*N* = 194) and HIV-negative women (*N* = 2116). Hence, five cohorts were included in the meta-analysis. The designs had similarities: women were included in the second or third trimester and were randomised 1:1 to IIV and a control group. In the RCTs from Bangladesh and Mali, the control group received another vaccine, while in the RCTs from Nepal and South Africa, the control group received a saline placebo. Children were followed to ~6 months of age. The main outcomes were related to influenza, and as shown in several meta-analyses, IIV protected mother and child partially from influenza-confirmed illness, with effect estimates ranging from 27% to 70%. All RCTs were assessed as low risk of bias (Appendix A); all were randomised trials with allocation concealment and intention-to-treat analyses, and in all trials, at least the assessors of outcomes, if not the actual vaccinators, were blinded (Table 1).

### 3.1. Maternal Outcomes

There were few maternal deaths in the RCTs, and even fewer from presumed infectious causes; no death was due to influenza in either the intervention or the control groups. Maternal IIV was associated with a risk ratio (RR) of dying from all causes of 1.48 (0.52–4.16) (Figure 1); for infectious causes of death, there were three deaths in the IIV groups (one in Mali, and two in South African HIV positive and negative, respectively) and no deaths in the control groups (*p* = 0.25).

There was a total of 37 non-influenza infectious adverse events in the IIV groups vs. 18 events in the control groups. This yielded a twofold higher risk of infectious adverse events from the time of vaccination to the end of follow-up (RR = 2.01 (1.15–3.50) (Figure 2)). The effect estimates from all four trials were above 1, ranging from 1.10 (0.38–3.14) in HIV-positive South African women to 2.97 (0.96–9.19) in Mali (Figure 2). Data were not available from Nepal. In the two cohorts from South Africa using a saline placebo, the RR was 1.72 (0.88–3.36) (data not shown).

### 3.2. Child Outcomes

There was no information on the sex of the children in any of the RCTs.

The effect estimates for miscarriage and stillbirth associated with maternal IIV had large confidence intervals and varied considerably. The meta-estimate was 1.06 (0.78–1.44) (Appendix A). Likewise, the effect estimates for infant mortality had large confidence intervals and varied considerably. The meta-estimate for infant mortality was 1.11 (0.87–1.41) (Appendix A). If the analysis was limited to presumed infectious deaths, the estimate was 0.94 (0.53–1.68) (Appendix A). In a combined analysis of miscarriage/stillbirth/infant death, the meta-estimate was 1.09 (0.90–1.31) (Figure 3).

IIV was associated with a higher risk of child adverse events/hospitalisations for infectious diseases. The results from the three RCTs that reported this outcome were homogeneous, showing effect estimates between 1.22 and 1.54. In total, there were 213 events in the IIV group vs. 154 in the control groups. The meta-estimate was 1.36 (1.12–1.67) (Figure 4). In the two South African cohorts using a saline placebo, the RR was 1.27 (0.96–1.69) (data not shown). Limiting the analysis to infections reported within the neonatal period, the estimate was 1.59 (1.12–2.27) (Appendix A).

## 4. Discussion

In meta-analyses of the existing four RCTs, all conducted in low-income settings, IIV in pregnancy had no effect on all-cause mortality of women and infants and was associated with a twofold higher risk of non-influenza infectious adverse events in women and with a 36% higher risk in their offspring up to 6 months after delivery.

### 4.1. Strengths and Weaknesses

The analysis is based on RCTs; thus, the risk of bias and confounding, particularly in the form of ‘healthy vaccinee bias’ or ‘frailty bias’, was limited. The RCTs were observer blinded; hence, the registration of deaths, hospital admissions, and other adverse events should not be affected by knowledge about the treatment. There was no indication of publication bias, but with four RCTs, the power to assess this aspect was limited.

Given the fact that vaccines have non-specific effects, the use of an active placebo in two of the RCTs hampers interpretation [28]. The use of pneumococcal vaccines in the Bangladesh trial could have provided benefits that confounded the comparison with IIV [29], and it has also been suggested that the administration of meningococcal vaccines to mothers in the Mali trial could have conferred benefits to the offspring [23]. However, effect estimates appeared similar in the cohorts using a saline placebo and those using a comparator vaccine.

A clear limitation is that the RCTs were not designed to study non-specific effects [10]. We relied on the reporting of adverse events. They might not have been reported consistently. The reporting of hospital diagnoses in Mali and South Africa was carried out per diagnosis, not per individual. Hence, a child could have several diagnoses as well as several admissions. However, only a few had several diagnoses/admissions (see the Methods Section) [26,27], and the lack of precision in the number of hospitalisations for infections should be similar for intervention and control groups.

A further limitation is the relatively small number of RCTs and the fact that all the RCTs were conducted in resource-limited settings—namely, Bangladesh, South Africa, Mali, and Nepal. All four are relatively poor nations with inadequate public health systems and generally poor living conditions that may exacerbate adverse reactions from IIV. Thus, the results might not be extrapolated to other scenarios with a higher quality of care in the pre- and perinatal stages of life. Furthermore, apart from the HIV-positive cohort, only healthy women were included in the RCTs. Only PubMed and Embase databases were searched for records, whereas Cochrane Library and grey literature were not searched; however, we do anticipate that any RCT of IIV would have been published in a PubMed indexed journal.

### 4.2. Interpretation

The RCTs showed convincingly that IIV was associated with a substantially lower risk of influenza in mother and child. Uncomplicated influenza in otherwise healthy adults and children is rarely fatal, but it can be highly disruptive, and recovery can be prolonged, with persistent fatigue and malaise that can last for weeks after the immediate infection is over [30]. Influenza can initiate immunosuppressive mechanisms, creating an ideal environment for opportunistic pathogens to grow out and induce co-infections, and influenza can be complicated by pneumonia [31]. The South African RCT reported that vaccination with IIV during pregnancy reduced the risk for acute lower respiratory infection hospitalisation by 58% during the first 90 days of life [32]. A meta-analysis of three RCTs conducted in Nepal, Mali, and South Africa showed that IIV was associated with significant reductions in severe clinical pneumonia during the influenza season [33]. Furthermore, the Nepal trial reported that during the influenza season, the proportion of infants who were small for gestational age was lower, and the mean birth weight was higher in the influenza vaccine group than in the control group [34].

By preventing influenza infection, IIV undoubtedly has prevented many negative direct and indirect consequences. However, in spite of these beneficial effects of IIV, here we report that the RCTs consistently found more non-influenza infectious adverse events among IIV recipients and their offspring. The meta-estimates indicated a significantly higher risk of non-influenza infectious adverse events among both pregnant women and children in these low-income settings.

All cases of influenza and severe pneumonia were registered during the trials, but other infections were apparently only registered if they were severe enough to lead to hospital admission. Hence, it is not possible to directly compare the beneficial effect on influenza and severe pneumonia with the observed negative effect on non-influenza infections. In absolute numbers, there were many more cases of influenza and severe pneumonia prevented by IIV than there were excess non-influenza adverse events. However, the lack of effect on all-cause mortality could indicate that the benefits for mortality of being protected against influenza might have been balanced by the higher incidence of severe non-influenza adverse events in the vaccinated groups.

It adds credibility to the findings that other non-live vaccines—namely, DTP vaccines [14], hepatitis B vaccines [15], inactivated polio vaccines [16], pentavalent vaccines [17], and new malaria vaccines [35] have been associated with negative non-specific effects in females, i.e., in spite of providing protection against the target disease, the vaccines were associated with higher all-cause mortality, primarily due to a higher risk of non-target infectious diseases. Furthermore, a campaign with the H1N1 influenza vaccine was recently found to be associated with higher female versus male mortality [19,21]. These results are primarily from observational studies since it was not possible to conduct RCTs of these already recommended vaccines. However, the typical biases in vaccine studies (‘healthy vaccinee bias’, ‘frailty bias’) favour the vaccinated, and therefore, observational studies showing negative effects of a vaccine on overall mortality are particularly noteworthy. Furthermore, we used the knowledge about non-live vaccines to predict that an RCT of a new malaria vaccine would show negative effects on overall mortality in females [36,37]. The consistency of the findings and the ability to predict future events limit the possibility that the present findings of an increased risk of non-influenza infectious adverse events associated with IIV are caused by bias.

### 4.3. Immunological Mechanisms

Immunological studies have shown that live vaccines, via epigenetic modulations, can induce innate immune training [37], whereas non-live vaccines induce innate tolerance [38,39]. In an experiment testing the effect of BCG on the response to IIV, BCG exerted an overall immunostimulatory effect measured on in vitro cytokine production to unrelated stimuli, but while IIV resulted in enhanced TNF-α production upon stimulation with LPS and C. albicans, production of IL-1β was decreased upon stimulation with *C. albicans*, *S. aureus* and *M. tuberculosis* [40]. Furthermore, in a recent study, IIV plus BCG resulted in increased spontaneous in vitro IL-6, TNF-α, and IL-1β production, and IIV alone led to increased IL-1R and IL-6 responses after SARS-CoV-2 or LPS restimulation; however, IIV alone had no effect on spontaneous cytokine production, and contrary to BCG, IIV had no effect on TNF-α production to LPS [Reference: https://www.medrxiv.org/content/10.1101/2020.10.14.20212498v1, accessed on 7 December 2021]. Taken together, these experiments show that IIV induces less innate immune training than that seen for live vaccines such as BCG. Some results indicated that IIV may induce some degree of innate tolerance; such effects may be particularly important in low-resource settings with a high burden of infectious diseases.

## 5. Conclusions

Health authorities justify IIV immunisation during pregnancy to reduce all-cause mortality and morbidity in mother and child. However, in spite of protecting against influenza in mother and child, there was no indication that IIV protected against all-cause mortality.

Similar to other non-live vaccines, IIV was associated with an increased risk of unrelated infections. Since data on influenza morbidity were collected meticulously, whereas data on non-related infections were only available through the adverse events registration, and thus represented more severe and rare events, it was not possible to directly compare the clear beneficial effects of IIV on influenza with the potential negative non-specific effect on the risk of other infections.

Nonetheless, the estimates in relation to overall mortality suggest that there is sufficient equipoise to conduct further RCTs in pregnant women. Coverage for this risk group is generally quite low worldwide; therefore, robust evidence is needed to enhance vaccine uptake in pregnant women. To be confident about the overall benefits of routine use of IIV in low-resource settings, further large RCTs with a true placebo and which include measurements of non-specific outcomes are needed before IIV can be made policy in these settings.

## Figures and Tables

**Figure 1 vaccines-09-01452-f001:**
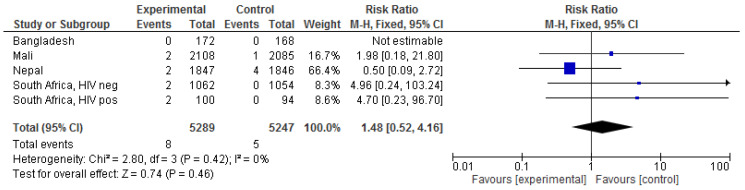
The effect of influenza vaccination in pregnancy on maternal mortality overall (excluding accidents/suicide). In the Bangladesh trial, a maternal death due to anaesthesia complications in the control group was judged an accident. In the Mali trial, Figure 1 presents 4 vs. 2 deaths, but the text and the appendix state 2 vs. 3 deaths. Two deaths due to electrocution, and ‘cervical fracture, post-trauma’ in the control group were excluded. In the Nepal trial, the abstract indicates that 5 vs. 3 women died in the control and intervention groups, respectively. The text and the Appendix A indicate 5 vs. 2, and these numbers were used for this study. Maternal death due to suicide in the control group in the Nepal trial was excluded.

**Figure 2 vaccines-09-01452-f002:**
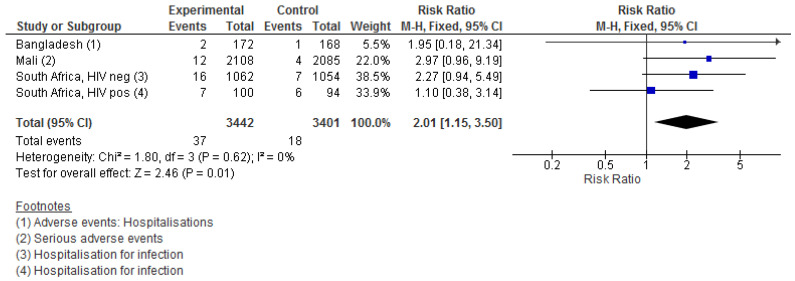
The effect of influenza vaccination in pregnancy on maternal non-influenza infectious adverse events. In Bangladesh, this was reported as hospitalisations. The following were considered infectious: fever; appendicitis; diarrhoea. In Mali, this was reported as serious adverse event. The following were considered infectious: chorioamnionitis; serious infection in pregnancy; peritonitis. In South Africa, this was reported as hospitalisations for infections. Among HIV-positive cohort, hospitalisation due to TB in the control group was excluded, as it was assumed as being pre-existing.

**Figure 3 vaccines-09-01452-f003:**
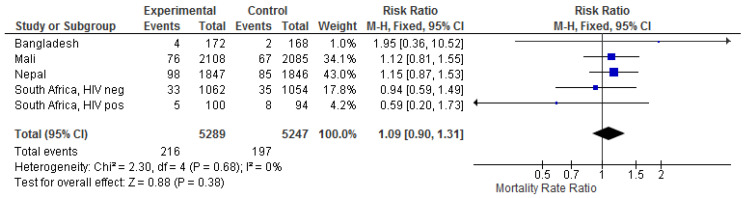
The effect of influenza vaccination in pregnancy on overall miscarriage/stillbirth/infant mortality.

**Figure 4 vaccines-09-01452-f004:**
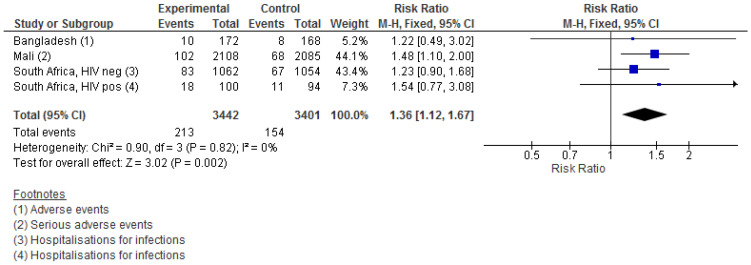
The effect of influenza vaccination in pregnancy on infant non-specific infectious disease adverse events. In Mali and South Africa, infection groups are not mutually exclusive; hence, an ill child could contribute to several groups at the same time. In Mali, the following diagnoses were assumed infectious: neonatal infection; respiratory infection; malaria; meningitis; gastrointestinal infection; unspecified infection; bacteraemia. In South Africa, in HIV-negative cohort, one case of sepsis <3 days in each group was assumed to be included in the cases of sepsis <28 days and therefore not counted twice.

**Table 1 vaccines-09-01452-t001:** Randomised controlled trials of influenza vaccination in pregnancy.

Country	Author, Year	Number of Pregnant Women	Influenza Vaccine, Type	Control Group Treatment	Time of Randomisation and Vaccination	Follow-Up	VE against Laboratory-Confirmed Influenza in Women	VE against Laboratory-Confirmed Influenza in Infants	Risk of Bias
Bangladesh	Zaman et al. 2008 [24]	340	3-valent IIV	23-valent pneumococcal polysaccharide vaccine	3rd trimester	24 weeks of age	N/A	63% (5–85%)	Low: Randomisation, allocation concealment, intention-to-treat analysis, observer blinding.
South Africa	Madhi et al. 2014 [27]	HIV neg: 2116HIV pos: 194	3-valent IIV	Saline	2nd + 3rd trimester (weeks 20–36)	24 weeks of age	HIV neg: 50% (15–71%) HIV pos: 58% (0–82%)	HIV neg: 49% (12–70%) HIV pos: 27%; *p* = 0.60	Low: Randomisation, allocation concealment, intention-to-treat analysis, observer blinding.
Mali	Tapia et al. 2016 [26]	4193	3-valent IIV	4-valent meningococcal vaccine	3rd trimester (≥28 weeks)	6 months of age	70% (42–86%)	33% (4–54%)	Low: Randomisation, allocation concealment, intention-to-treat analysis, observer blinding.
Nepal	Steinhoff et al. 2017 [25]	3693	3-valent IIV	Saline	2nd + 3rd trimester (weeks 17–34)	6 months of age	31% (−10–56%)	30% (5–48%)	Low: Randomisation, allocation concealment, intention-to-treat analysis, observer blinding.

IIV = inactivated influenza vaccine; VE = vaccine efficacy.

## Data Availability

Not applicable.

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
