# Peer review of "Does Influenza Vaccination during Pregnancy Have Effects on Non-Influenza Infectious Morbidity? A Systematic Review and Meta-Analysis of Randomised Controlled Trials"

_vaccines, 2021, doi:10.3390/vaccines9121452_

Round 1
Reviewer 1 Report
Does influenza vaccination during pregnancy have effects on 2 non-influenza infectious morbidity? A systematic review of 3 randomised trials Has been reviewed.
Influenza infection in pregnancy has been associated with increased severity of influenza infection, both in developed and developing countries. This review takes into account only RCTs carried out in developing countries and thus might not be extrapolated to other scenarios with higher quality of care in the pre and perinatal stage of life.
Influenza vaccination of pregnant women can increase protection against influenza and complications derived from influenza in the newborn for this reason WHO recommends pregnant women and especially those that deliver in the epidemic season, to use inactivated influenza vaccine. Coverage for this risk group is generally quite low worldwide, therefore robust evidence is needed to enhance vaccine uptake in pregnant women.
The authors describe the results of 4 RCTs, which is far from a broad review based on the lack of such studies, yet this should not undermine other studies that have proven VE to prevent ICU admission, for example, during pregnancy. Positive evidence should be delivered in the discussion section to stress the importance of Influenza vaccination. Severe infection/disease leads to the increased incidence of secondary infections, probably through immune-suppression. Beumer MC, Koch RM, van Beuningen D, et al. Influenza virus and factors that are associated with ICU admission, pulmonary co-infections and ICU mortality. J Crit Care. 2018;50:59-65.
In all it is a well written paper to which there are a few inquiries to be cleared.
This sentenece is confusing on page 2 lines 49-50
These effects risk 48 being overlooked in vaccine trials as study protocols are usually only designed to find 49 pathogen-specific effects.
Please rephrase so to make it understandable.
Line 51 and line 54
By measles vaccine the authors are referring to MV containing vaccine or directly to MMR vaccine?
Line 56
Why do you affirm that the results of studies assessing non live vaccine is wrong?
Line 70
What do the authors means by “; it yielded no effect of IIV”. The vaccine had no effectiveness in protecting from the infecton or had no apparent risk of enhancing other infections? ^Pleas e clarify.
Page 3 Methods section Line 132
“As the denominator, for simplicity, we chose number of randomised pregnant women for all outcomes.” Do you mean the total number of women included in RCTs ?
Author Response
Influenza infection in pregnancy has been associated with increased severity of influenza infection, both in developed and developing countries. This review takes into account only RCTs carried out in developing countries and thus might not be extrapolated to other scenarios with higher quality of care in the pre and perinatal stage of life.
CB: The point is well taken. We have emphasised that in the discussion “A further limitation is that all the RCTs were conducted in resource-limited settings, and thus the results might not be extrapolated to other scenarios with higher quality of care in the pre- and perinatal stage of life.”
Influenza vaccination of pregnant women can increase protection against influenza and complications derived from influenza in the newborn for this reason WHO recommends pregnant women and especially those that deliver in the epidemic season, to use inactivated influenza vaccine. Coverage for this risk group is generally quite low worldwide, therefore robust evidence is needed to enhance vaccine uptake in pregnant women.
CB: We very much agree. We have emphasised that in the discussion ”Coverage for this risk group is generally quite low worldwide, therefore robust evidence is needed to enhance vaccine uptake in pregnant women”.
The authors describe the results of 4 RCTs, which is far from a broad review based on the lack of such studies, yet this should not undermine other studies that have proven VE to prevent ICU admission, for example, during pregnancy. Positive evidence should be delivered in the discussion section to stress the importance of Influenza vaccination. Severe infection/disease leads to the increased incidence of secondary infections, probably through immune-suppression. Beumer MC, Koch RM, van Beuningen D, et al. Influenza virus and factors that are associated with ICU admission, pulmonary co-infections and ICU mortality. J Crit Care. 2018;50:59-65.
CB: We already have a long section in the discussion that emphasises the negative consequences of influenza infection. We have added that “Influenza can initiate immunosuppressive mechanisms, creating an ideal environment for opportunistic pathogens to grow out and induce co-infections.” And “By preventing influenza infection, IIV undoubtedly has prevented many negative direct and indirect consequences”
In all it is a well written paper to which there are a few inquiries to be cleared.
CB: Thank you.
This sentenece is confusing on page 2 lines 49-50
These effects risk being overlooked in vaccine trials as study protocols are usually only designed to find pathogen-specific effects.
Please rephrase so to make it understandable.
CB: The sentence has been revised. “The current system for testing vaccines is not set up to assess non-specific effects, as it usually focusses on pathogen-specific effects”
Line 51 and line 54
By measles vaccine the authors are referring to MV containing vaccine or directly to MMR vaccine?
CB: We have clarified that we talk about measles-containing vaccine (i.e. both measles vaccine and MMR vaccine)
Line 56
Why do you affirm that the results of studies assessing non live vaccine is wrong?
CB: We are not sure we understand; we did not affirm anything about studies being wrong.
Line 70
What do the authors means by “; it yielded no effect of IIV”. The vaccine had no effectiveness in protecting from the infecton or had no apparent risk of enhancing other infections? ^Pleas e clarify.
CB: We have now clarified “A pooled analysis of two RCTs of IIV to pregnant women with respect to maternal and child all-cause mortality was subsequently published; it yielded no effect of IIV on these outcomes”
Page 3 Methods section Line 132
“As the denominator, for simplicity, we chose number of randomised pregnant women for all outcomes.” Do you mean the total number of women included in RCTs ?
CB: Yes. We have specified: “we chose number of randomised pregnant women included in the RCTs for all outcomes”
Reviewer 2 Report
Please review the comments below:
1) Did you follow the PRISMA guidelines? A PRISMA flow diagram is much needed. It is a must when doing a systematic review. I am not seeing the checklist.
2) It seems that meta-analysis was done, but the title does not mention it.
3) Inclusion and Exclusion criteria are extremely vague. Did you look at articles in English only? Why did you only look at the two databases -- How about the Cochrane Library?
4) How about a Quality assessment of the articles?
5) I am not seeing Appendix 1; however, did you use any Mesh terms when searching within PubMed?
6) Did you find any strengths or limitations?
7) Extensive English proofreading is needed.
Author Response
1) Did you follow the PRISMA guidelines? A PRISMA flow diagram is much needed. It is a must when doing a systematic review. I am not seeing the checklist.
CB: We apologise, we had forgot to upload. We have now inserted the PRISMA flow diagram as supplementary figure 1. We also enclose the PRISMA checklist.
2) It seems that meta-analysis was done, but the title does not mention it.
CB: We have now added “and meta-analysis” in the title.
3) Inclusion and Exclusion criteria are extremely vague. Did you look at articles in English only? Why did you only look at the two databases -- How about the Cochrane Library?
CB: We have now clarified. We did not include the Cochrane Library. We have added this as a limitation to the review. We did not exclude any non-English articles, but only identified articles in English language.
4) How about a Quality assessment of the articles?
CB: We write that “All RCTs were assessed as low risk of bias; all were randomised trials with allocation concealment, intention-to-treat analyses, and in all trials, at least the assessors of out-comes, if not the actual vaccinators, were blinded (Table 1)”.
5) I am not seeing Appendix 1; however, did you use any Mesh terms when searching within PubMed?
CB: Sorry, appendix 1 is now added.
6) Did you find any strengths or limitations?
CB: We have a section on strengths and limitations in the discussion section.
7) Extensive English proofreading is needed.
CB: We have done our best.
Reviewer 3 Report
This meta-analysis by Hansen et al. reveals non-specific side effects (NSEs) from inactivated influenza vaccine (IIV) administered to pregnant mothers in four randomized control trials (RCTs). The analysis was thorough, paying attention to multiple variables within the limitations of available reports. Data collection and extraction were unbiased and the authors noted all reported adverse effects, including mother and infant mortality, miscarriage, and non-influenza infection events. There are no major weaknesses in the analysis. The studies should be useful in re-evaluating how the vaccine recipients are advised. My only concern is the relatively small number of RCTs in only four countries, namely Bangladesh, South Africa, Mali and Nepal. All four are relatively poor nations with inadequate public health system and a generally poor living conditions that may exacerbate adverse reactions from IIV. Did the authors find any RCT in the affluent nations such as the industrialized G8 group? If so, the results should be added or discussed.
Author Response
We thank the reviewer for these positive comments. We found no RCTs in more affluent countries. We have clarified that, and we emphasise the consequences for the extrapolations that can be made (see also response to reviewer 1). The section now reads “A further limitation is the relatively small number of RCTs and the fact that all the RCTs were conducted in resource-limited settings, namely Bangladesh, South Africa, Mali and Nepal. All four are relatively poor nations with inadequate public health systems and generally poor living conditions that may exacerbate adverse reactions from IIV. Thus, the results might not be extrapolated to other scenarios with higher quality of care in the pre and perinatal stage of life, limiting the generalisability to high-income settings”
Round 2
Reviewer 2 Report
1) Risk of Bias assessment ought to be performed
2) PRISMA diagram needs to have straight lines
3) I am not still seeing the complete list of search strategies for PubMed and EMBASE.
4) Title needs to say randomized controlled trials, as you have mentioned that RCTs were reviewed.
Author Response
Below, please find a point-to-point response to these additional comments.
1) Risk of Bias assessment ought to be performed
CB: It has now been inserted as Table S2.
2) PRISMA diagram needs to have straight lines
CB: Corrected.
3) I am not still seeing the complete list of search strategies for PubMed and EMBASE.
CB: It was included as appendix 1. It has now additionally been inserted under the PRISMA diagram.
4) Title needs to say randomized controlled trials, as you have mentioned that RCTs were reviewed.
CB: We have included the term in the title.